# Childhood Hearing Impairment in Senegal

**DOI:** 10.3390/genes14030562

**Published:** 2023-02-23

**Authors:** Yacouba Dia, Birame Loum, Yaay Joor Koddu Biigé Dieng, Jean Pascal Demba Diop, Samuel Mawuli Adadey, Elvis Twumasi Aboagye, Seydi Abdoul Ba, Abdoul Aziz Touré, Fallou Niang, Pierre Diaga Sarr, Cheikh Ahmed Tidiane Ly, Andrea Regina Gnilane Sène, Carmen De Kock, Rhiyana Bassier, Kalinka Popel, Rokhaya Ndiaye Diallo, Ambroise Wonkam, Bay Karim Diallo

**Affiliations:** 1Division of Human Genetics, Faculty of Medicine, Pharmacy and Odontology, University Cheikh Anta Diop (UCAD), Dakar 10700, Senegal; 2Department of Oto-Rhino-Laryngology, Albert Royer Children’s Hospital, Dakar 10700, Senegal; 3Department of Neonatology, Albert Royer Children’s Hospital, Dakar 10700, Senegal; 4Division of Human Genetics, Faculty of Health Sciences, University of Cape Town, Cape Town 7925, South Africa; 5West African Centre for Cell Biology of Infectious Pathogens (WACCBIP), University of Ghana, Legon, Accra P.O. Box LG 54, Ghana; 6McKusick-Nathans Institute and Department of Genetic Medicine, Johns-Hopkins University School of Medicine, Baltimore, MD 21205, USA

**Keywords:** hearing impairment, childhood, causes, consanguinity, Senegal, Africa

## Abstract

We recently showed that variants in *GJB2* explained Hearing Impairment (HI) in 34.1% (*n* = 15/44) of multiplex families in Senegal. The present study aimed to use community-based nationwide recruitment to determine the etiologies and the clinical profiles of childhood HI in Senegal. Participants with early onset HI were included after clinical examination, including audiological assessment by pure tone audiometry and/or auditory brainstem response. We investigated a total of 406 participants from 295 families, recruited from 13/14 administrative regions of Senegal. Male/female ratio was 1.33 (232/174). Prelingual HI was the most common type of HI and accounted for 80% (*n* = 325 individuals). The mean age at medical diagnosis for congenital HI was computed at 3.59 ± 2.27 years. Audiological evaluation showed sensorineural HI as the most frequently observed HI (89.16%; *n* = 362 individuals). Pedigree analysis suggested autosomal recessive inheritance in 61.2% (63/103) of multiplex families and sporadic cases in 27 families (26.2%; 27/103), with a consanguinity rate estimated at 93% (84/90 families). Genetic factors were likely involved in 52.7% (214/406) of the cases, followed by environmental causes (29.57%; 120/406). In 72 cases (17.73%), the etiology was unknown. Clinically, non-syndromic HI was the most common type of HI (90.6%; *n* = 194/214 individuals). Among families segregating syndromic cases, type 2 Waardenburg syndrome was the most common (36.3%; 4/11 families). This study revealed putative genetic factors, mostly associated with high consanguinity rate, as the leading causes of early-onset HI in Senegal. The high consanguinity could provide a good opportunity to identify variants in known and novel genes involved in childhood HI.

## 1. Introduction

Of all the sensory deficits worldwide, hearing impairment (HI) remains the most disabling, with the highest rate for age-standardized disability of life [1]. According to the World Health Organization (WHO), over 466 million people (5% of the world’s population) are living with HI, of whom 34 million are children [2]. It is expected that by 2050 nearly 2.5 billion people will experience some degree of HI [3]. Without early detection and intervention, HI can cause detrimental impacts on speech, language development, educational and cognitive outcomes in children [4,5]. The prevalence of childhood HI (CHI) varies widely from one geographic region to another [6,7,8,9], with a global prevalence ranging from 1.7 to 3.6/1000 in developed countries [7,10,11,12]. In the developing settings, particularly in sub-Saharan Africa (SSA), permanent CHI affected up to 1 in 100 children [8]. This disparity could be attributed to the distribution of the risk factors that vary significantly between high- and lower-income countries. In the high-income countries, more than 50% of congenital HI are due to genetic etiologies, while in the lower-income countries, specifically in SSA, environmental factors remain a major cause of CHI, with cerebral spinal meningitis being the most prevalent in the African Meningitis Belt [13,14,15]. Despite the notable progress made to address the rising prevalence of CHI by implementing vaccine programs with widespread coverage and prenatal follow-up care visits, the overall picture of CHI continues to worsen in SSA [8]. This suggests, like in most countries in epidemiologic transition, that non-environmental factors, and specifically genetics factors, may significantly contribute to the upward trend seen in estimates of the global prevalence of CHI in Africa [8].

In previous studies from numerous SSA countries, data showed that genetic factors contribute from 30% to 50% of CHI cases [16]. CHI due to genetic factors can be isolated (non-syndromic HI (NSHI)) or associated with other organ abnormalities (syndromic HI). Roughly 30% of all congenital HI are syndromic, with over 400 syndromes described to date [17]. NSHI is the most common type of HI, accounting for 70% of all hereditary HI types. Among NSHI, nearly 80% of cases are inherited in autosomal recessive mode [18], characterized by being profound and pre-lingual. It is well known that consanguineous mating increases the risk of autosomal disorders in the offspring [19]. In west Africa, so far the highest consanguinity rate (80%) has been reported in Senegal [20]. Furthermore, Demographic and Health Survey data revealed that the consanguinity rate varies from 50% to 80% depending on the ethnic group, with the highest rates prevailing among Fulani, Serer and Wolof ethnic groups [20].

Assessing the etiological profile of CHI requires a consistent methodology and population-based recruitment. To the best of our knowledge, only one retrospective study has attempted to determine the etiological profile of HI among 178 children in one out of fourteen administrative regions of Senegal [21]. Environmental factors were identified as the leading causes of HI (93%), while the etiology was unknown in 7% of the cases [21]. Moreover, there is not yet a universal newborn hearing screening program in Senegal. We recently showed that variants in *GJB2* explained HI in 34.1% of the cases (*n* = 15/44) in a select group of multiplex families in Senegal [22]. In order to get the full spectrum of CHI-associated etiologies across the country, we extended our recruitment to determine the clinical characteristics of CHI in Senegalese individuals living with HI.

## 2. Materials and Methods

### 2.1. Ethical Approval Statement

The present study was conducted according to the guiding principles of the Declaration of Helsinki. We applied and obtained ethical approval (CER/UCAD/AD/MSN/034/2020) from the Research Ethics Committee of the University Cheikh Anta Diop, Dakar, Senegal, and the University of Cape Town, Faculty of Health Sciences Human Research Ethics Committee (HREC 104/2018). The scope of the study was clearly explained to the participants who were ≥18 years of age and the parents/guardians in the case of minors (participants <18 years), in their fluent language. Informed consent was obtained from the participants (≥18 years of age) or their parents/guardians (for participants <18 years of age), including for publishing their photographs, when applicable.

### 2.2. Operational Definitions

In the context of the present study, HI was defined as: (1) acquired, if an environmental factor was identified as the cause of HI; (2) likely genetic, when there is segregation of the condition in the family with at least two affected individuals and without an obvious environmental factor, in the sporadic case from a consanguineous mating without a known environmental factor, or in case of a well clinically defined syndrome; (3) unknown, when the cause was not established as environmental or genetic, as previously reported in Cameroon [13].

### 2.3. Study Settings, Populations, and Procedures

The present study was performed at the Division of Human Genetics, Faculty of Medicine, Pharmacy and Odontology of Cheikh Anta Diop University, Dakar, Senegal, and at the department of Otorhinolaryngology of the children hospital, Albert Royer, Dakar, in collaboration with the Division of Human Genetics, Faculty of Health Sciences, University of Cape Town, Cape Town, South Africa. We performed an observational study by recruiting participants from thirteen out of fourteen administrative regions of Senegal. There is only one public school for the deaf in the country, i.e., Centre Verbo-Tonal (CVT) de Dakar. The schools for the deaf in the other administrative regions are not funded by the Senegalese government. Therefore, outside Dakar, participants were recruited from non-public schools as well as through community engagement. Only participants with HI greater than 25 decibels (dB) that started before the age of 15 years were enrolled. In the administrative region of Dakar, hearing-impaired participants were recruited from the Albert Royer National Children’s Hospital (Otorhinolaryngology and Neonatology departments), and the school for the deaf, Centre Verbo-Tonal (CVT) de Dakar. In the other administrative regions, we recruited participants from schools for the deaf and/or in the community. For the community-based recruitment, the recruitment strategy involved identifying a local professional and/or community member, who took us to relevant families. In most regions, we started our recruitment with one community leader. However, over the course of the recruitment process, we identified additional community leaders who actively participated in the process of identifying families. Because these community leaders were the pillars of community-based recruitment, in absence of one in the Sédhiou administrative region, we could not perform any recruitment in that region (Figure 1).

After signing informed consent, all participants and the parents/guardians were interviewed for family and medical history, including exposure to prenatal, perinatal, and postnatal risk factors. Medical records were reviewed by a general practitioner, a medical geneticist, and/or an oto-rhino-laryngologist where applicable. All relevant data were collected through a questionnaire.

For suspected syndromic HI, based on the clinical assessment or family history, additional tests were performed when applicable, including thyroid hormones (FT4, T3, and TSH), glycemia, serum creatinine level, kidney and thyroid gland ultrasounds, electrocardiography, and/or fundus examination, to refine the diagnosis as described previously in Mali [14]. For familial or sporadic cases with a putative genetic origin, peripheral blood samples were taken when possible and three-generation pedigrees were drawn using Progeny software (Progeny Genetics LLC, Delray Beach, FL, USA).

### 2.4. Audiological Evaluation

An otoscopic examination was carried out for all the participants to detect probable abnormalities in the outer or middle ear, and the cerumen plug was removed when applicable, before audiological evaluation. We performed an age-appropriate audiological examination. Pure Tone Audiometry (PTA) was performed with a mobile audiometer (250 Hz to 8000 Hz) (KUDUWAVE^TM^ N°0901-04011, Cape Town, South Africa), for participants aged 2 years and older. In the schools for the deaf, PTA tests that were established before admission, were reviewed for some participants to detect progressive hearing loss. The hearing threshold was calculated as the average hearing level at 0.5, 1.0, 2.0, and 4.0 kHz. Normal hearing was defined as hearing thresholds less than 25 dB.

For patients having a conductive or mixed HI, tympanometry was performed to confirm the mechanism of HI, when possible.

For the participants who were too young (<2 years) for a PTA testing, an otoacoustic emissions (OAE) test was performed followed by an auditory brainstem response (ABR) test, when possible.

### 2.5. Data Analysis

Descriptive statistics were performed using R software version 4.0.3 (R core team 2020, Vianna, Austria).

## 3. Results

### 3.1. Participant Demographics

A total of 406 participants, belonging to 295 families, were recruited from 13 out of 14 administrative regions of Senegal. Sédhiou was the only administrative region that was not covered since we have not identified a local collaborator willing to assist with community recruitment in that region. The majority of the families (75.3%; *n* = 222/295) were recruited from the western administrative regions (Dakar, Thiès, and Diourbel) (Figure 1).

Prelingual HI was the most common type of HI (77.6%; *n* = 316/406), followed by postlingual HI (12.81%; *n* = 52/406). Congenital HI accounted for 70.67% of HI (*n* = 287/406), with a mean age at medical diagnosis of 3.59 ± 2.77 years. The male/female ratio was 1.33 (232/174). The participants’ mean age at recruitment was 12 ± 9 years (range = 1–69 years). Formal primary education attainment was observed in most participants (44%; *n* = 179/357), while 14.49% (*n* = 71 individuals) had no formal education (Table 1).

### 3.2. Audiometric Characteristics and Management of Permanent HI

Of the 406 participants, 368 (90.6%) had undergone a PTA. For 38 patients of age under 2 years, OAE or ABR was conducted. Sensorineural HI was the most frequent type of HI observed and accounted for 89.2% of HI (*n* = 362/406). We classified the hearing levels according to the Global Burden Disease Hearing Loss Expert Group recommendation [23]. The majority of the participants (67.73%; *n* = 275/406) exhibited profound to total HI. HI was bilateral in 400 patients (98.7%) and symmetric in 380 patients (93.8%). Analysis of the audiometric curve pattern among 368 audiogram tests showed that 301 participants (81.8%) had a flat curve, while 13 participants (3.53%) exhibited cophosis (no curve) (Table 2).

In terms of HI rehabilitation, among 131 participants with moderate to severe HI, only 11.45% (*n* = 15) had an assistive hearing device. Of 275 participants with profound HI, who should benefit from cochlea implant(s) for hearing rehabilitation, only 3 participants (1%) were implanted (Table 2).

### 3.3. Etiologies of Childhood Hearing Impairment in Senegal

Putative environmental etiologies of HI accounted for 29.4% of participants (*n* = 120/406 cases). Chronic otitis media was the major environmental cause (9.38%; *n* = 38/406), followed by cerebral meningitis (6.16%; *n* = 25/406) (Table 3). In 72 participants (18%; *n* = 72/406 cases), the etiology of HI was unknown. The cause is likely genetic in 214 cases, (52.5%; *n* = 214/406), from 103 families. Pedigree analysis of 71 multiplex families (syndromic and non-syndromic HI) showed that autosomal recessive (AR) inheritance (Figure 2A) was the most frequently observed pattern of inheritance (88,73%; *n* = 63/71 families), followed by autosomal dominant (5.63%; 4/71) (Figure 2C) and recessive X-linked (4.23%; 3/71) (Figure 2B). The consanguinity rate ascertained in families segregating autosomal recessive and sporadic HI was set at 93% (84/90 families) (Appendix A).

Clinically, for patients with the suspected genetic origin, NSHI was the most common type of HI (90,65%; *n* = 194/214 cases). Twenty participants belonging to eleven families exhibited syndromic HI. We identified seven syndromes, with Waardenburg Syndrome (WS) being the most prevalent (36.3%, *n* = 4/11 families).

#### 3.3.1. Waardenburg Syndrome (WS)

Five cases of WS type 2, found in four families, were reported; three cases from two multiplex families (Figure 3D and Appendix A) and two sporadic cases from two simplex families (Figure 3A and Appendix A). For the multiplex family D (Figure 3D), the proband and his affected brother exhibited leukoderma and sapphire-blue eyes (Figure 3H) associated with profound sensorineural HI (Figure 3E,G), while the proband’s parents showed a normal phenotype. WS is known to be autosomal dominant, we, therefore, suspected an incomplete penetrance (Figure 3D).

#### 3.3.2. Labyrinthine Aplasia, Microtia, and Microdontia (LAMM) Syndrome

LAMM syndrome was suspected in a 7-year-old girl from a non-consanguineous marriage (Figure 4F), showing unilateral microtia (Type 3) (A), microdontia with wide-spaced teeth (B), periauricular skin tags (D), and asymmetrical face (E). The audiogram shows an asymmetrical hearing loss (profound hearing loss in the right ear and moderate in the left ear) (Figure 4).

#### 3.3.3. Pendred Syndrome

Four cases of Pendred syndrome, from a consanguineous family, exhibited profound sensorineural HI associated with euthyroid goiter (Figure 5).

#### 3.3.4. Diabetes-Deafness Syndrome

Three cases of Diabetes-deafness syndrome from a consanguineous marriage presented progressive post-lingual HI associated with diabetes-mellitus type 1 (Figure 6).

#### 3.3.5. Usher Syndrome

Three siblings with Usher syndrome type 2, without vestibular areflexia, were from a consanguineous mating (Figure 7). In addition to HI, these patients exhibited night vision impairment and constricted visual field. The ophthalmological examination showed the presence of retinitis pigmentosa.

#### 3.3.6. Down’s Syndrome

There was one case of Down’s syndrome, with a flat face, short neck, small ears, poor muscle tone, and profound sensorineural HI (Figure 8).

#### 3.3.7. Alport Syndrome

One case of Alport syndrome (a fourteen years old boy; III.6) showed profound sensorineural HI associated with renal failure (Figure 9).

#### 3.3.8. Uncharacterized Syndrome

Two cases with congenital HI had a non-specific syndrome characterized by profound HI, mental retardation, and developmental coordination disorder (Figure 10).

### 3.4. Consanguinity Rate among the Main Ethnic Groups

In 63 multiplex families segregating autosomal recessive HI (syndromic and non-syndromic HI), and 27 simplex families with NSHI, we estimated the consanguinity rate at 93% (*n* = 84/90 families). Fulani and Serer ethnic groups were more consanguineous compared to the Wolof ethnic group (Table 4).

### 3.5. Comparison of Our Results to Other Studies in the Neighboring Countries of Senegal

The etiological profile of CHI that we observed in our study differs from what is reported in the neighboring countries of Senegal (Gambia, Mali, and Guinea), except for Mauritania. In fact, environmental factors constitute the major cause of childhood HI in these countries, with cerebrospinal meningitis being the most prevalent environmental etiology in Mali (42%), Guinea (40%), and The Gambia (31%) (Table 5). Mauritania is the only neighboring country that shares two main specificities with Senegal regarding the etiological profile of CHI; the high consanguinity rate (61.3%) and the contribution of *GJB2* pathogenic variants (9.4%) in CHI (Table 5).

## 4. Discussion

The present study reports the main causes of CHI in Senegal based on an observational study, thereby constituting the most comprehensive study of the clinical characteristics and etiologies of HI among Senegalese children, to date. In fact, we recruited participants in 13/14 (93%) administrative regions to get a sample that fairly represents the hearing-impaired population of Senegal. Contrasting with reports from other SSA countries, where environmental factors remain a major cause of CHI, our study revealed a higher proportion of likely genetic etiology (52.7%), which is similar to reports from high-income settings [27,28]. This discrepancy might be due to the high consanguinity rate associated with a significant contribution of *GJB2* to NSHI in the Senegalese population [22]. Moreover, Senegal is in an epidemiological transition phase characterized by the reduction of infectious diseases [29]. In the last decade, Senegal has made notable progress; increasing the vaccine program implementation, prenatal visits, and childbirth. In fact, the meningitis W vaccine, MMR (Measles, Mumps, and Rubella) vaccine, and eight more vaccines are available free of charge in 84% of health facilities across the country [30]. Indeed, to reduce childbirth complications, in 2005, the national free delivery and cesarean policy was implemented across the country [31]. These interventions have significantly reduced the contribution of CHI-associated environmental risk factors.

Similar to other reports from SSA countries, in Mali [14], Ghana [32], and Cameroon [13], the sex ratio (1.33) is biased towards the male sex. The Joint Committee on Infants Hearing has not reported gender as a CHI-associated risk factor, however, it has been postulated that the anatomical and physiological differences between male and female auditory systems might potentially explain the bias in favor of the male sex [33]. In the context of limited resources, particularly in SSA, educating hearing-impaired girls are relegated to the back burner. Prelingual HI was the most common type of HI with a late diagnosis of congenital HI, which is consistent with the previous reports from SSA countries [13,14,32,34]. However, in the developed regions, HI is detected earlier. In the United States, the early hearing detection and intervention program screens more than 95% of all US newborns within the first month of life, which facilitates the diagnosis by the age of three months, and, when needed, an intervention within the first year of life [35]. To address the late screening and diagnosis, an association named DEGUE, which means hear in Wolof, was created in 2021 [36]. In a preliminary phase, systematic newborn hearing screening has been implemented in four maternity hospitals across four administrative regions of Senegal [36]. This may help to decrease the mean age at medical diagnosis. Despite the importance of modern technology for hearing rehabilitation, among 132 participants with moderate to severe hearing loss, only 15 (11.4%) had an assistive hearing device. This is consistent with the WHO reports in developing countries [2]. Poor access to assistive hearing devices and technologies could be due to the expensive cost of hearing aids rehabilitation [37]. In absence of early intervention, HI impedes speech and language development and can set affected children on a trajectory of sub-optimal educational attainment. In this study, we estimated the literacy rate at 47.8%, with the majority having a primary school level, as educational facilities for children living with HI are deficient in Senegal. There is only one public school for the deaf (Centre Verbo-Tonal, Dakar, Senegal), in the country.

Awareness regarding the CHI-associated risk factors and CHI’s impact on children’s well-being should be raised among communities. Postlingual HI in SSA is mainly caused by environmental factors. As previously reported in Ghana [32], Mali [14], and Cameroon [13], cerebrospinal meningitis, is the most common cause of postlingual HI. In this study, HI as sequelae of cerebral meningitis accounted for 6.2% (*n* = 25 cases). Senegal is located in the African “Meningitis belt”, characterized by seasonal outbreaks [38]. Since the introduction of the meningococcal W vaccine in Senegal, the incidence of cerebral meningitis has decreased [21]. Other environmental factors such as prematurity, cerebral malaria, and neonatal infections were also identified as etiologies of CHI in our cohort, but were not reported to play a significant role in HI in some SSA countries [13].

Among 63 multiplex families segregating autosomal recessive HI and 27 simplex families with NSHI, we report the highest consanguinity rate (93%) in SSA, to date [13,14,25,34,39]. Senegal is a culturally and ethnically diverse country, with twenty ethnic groups. Wolof, Fulani, and Serer constitute the main ethnic groups of Senegal and each one is socially stratified [40]. The Fulani ethnic group from the Northern region of Senegal (Toucouleur) remains the most socially stratified ethnic group with eleven hierarchical social classes [40]. Crossing caste boundaries within ethnic groups is challenging for lower social classes. In this case, consanguineous marriage remains the only option for lower social classes, which creates genetic isolates. In 1960, Pierre Catrelle predicted a social revolution among the Fulani ethnic group (Toucouleur) which could be supported by schooling, the development of communication means, feminization, and emigration to urban centers like Dakar [40]. Sixty-two years later, the caste system is still very much alive. The previous Ghanaian founder variant, *GJB2*: c.427C > T; p.(Arg143Trp), was identified at homozygous state in two families belonging to a lower Wolof social class (griot). However, we also found this variant at the compound heterozygous state in two other Wolof families, which means a marriage between different social classes. The Sufi brotherhood of the Layenne community, anchored in the Cap Vert peninsula (Dakar), attempts to break down caste barriers. Within the brotherhood, marriages between different social classes are tolerated. This should explain the compound heterozygous genotype observed in those families. Caste boundaries are not the only reason that might explain the higher consanguinity rate in Senegal. Another reason that encourages and sustains consanguineous marriage in Senegal, is the permanent quest for cohesion, marriage stability, and close ties between families.

Syndromic HI is phenotypically heterogeneous with over 400 syndromes described, to date. Among cases with the suspected genetic origin, we highlighted seven syndromes with WS type 2 being the most prevalent, as previously reported in Ghana and Mali [14,32]. We suspected LAMM syndrome in a 7 years old girl with a microtia and microdontia with wide-spaced teeth. *FGF3* screening and CT of the inner ear are needed to confirm this syndrome. NSHI was the most common type of genetic HI case [17], confirming what is already reported worldwide [41]. HI is genetically heterogeneous [42]. Since the first HI gene identification in 1995 [43], HI-associated genes are continuously being discovered [39], and some 1000 NSHI genes are still to be identified based on the inner ear transcripts [44]. The present study, illustrating a high consanguinity rate, suggests that Africa might be the next frontier of HI-associated gene discovery. Indeed, the use of Whole Exome Sequencing (WES) has allowed the identification of seven novel candidate genes among 51 families in Ghana [39]. In previous studies, including families within the present cohort, *GJB2* sequencing showed that almost 2/3 of investigated families (44 multiplex families) that are negative for *GJB2* variant are eligible for WES. This will allow the identification of variants in known genes and probably novel candidate genes that will further our understanding of hearing pathobiology.

A limitation of this study was the use of Pure Tone Audiometry (PTA) for preschool years (2–5 years), which was the only approach available. Despite that we performed behavioral audiometry using the same equipment as in children over 5 years (PTA), with an adaptation of the technique (conditioning with toys), this approach was time-consuming. In addition, depending on the child’s level of concentration for some children, the test was conducted over several days in order to obtain a reliable result. The present paper describes a comprehensive clinical profile of HI in an African country, i.e., Senegal, which will add to the scares and needed data from that region. The present data, showing a late age at diagnosis for CHI, support that in order to reduce the burden of CHI, policymakers must allocate appropriate resources by integrating universal newborn hearing screening into the health care system in Senegal. The high consanguinity rate equally urges the implementation of genome wide studies to refine the genetic cause of HI in families, opening an avenue for research opportunities.

## 5. Conclusions

In this study, we found putative genetic etiology in >50% of families, due to a high consanguinity rate (93%). National policy must be promoted in terms of raising awareness to discourage consanguineous marriages to decrease the occurrence of congenital HI in Senegal. A high consanguinity rate could provide an opportunity for novel HI gene discovery that will further our understanding of the molecular mechanism underlining HI.

## Figures and Tables

**Figure 1 genes-14-00562-f001:**
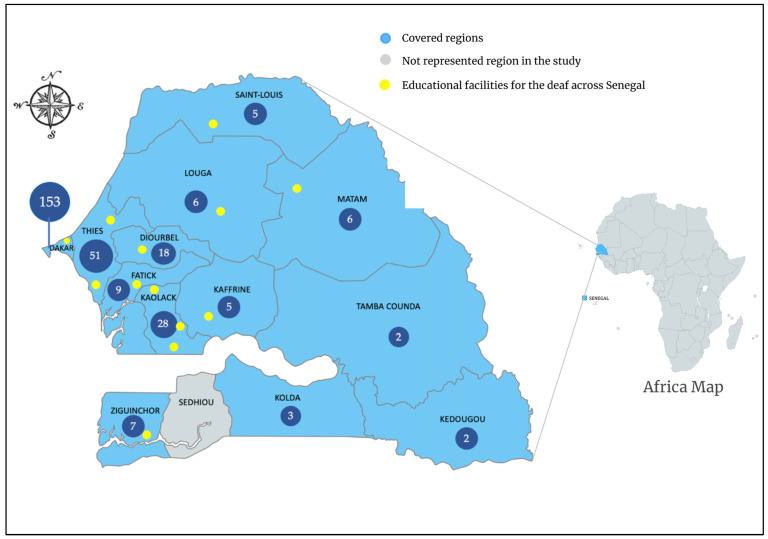
Regional distribution of 295 families living with HI in Senegal. More families were recruited from the Western regions, particularly in Dakar. The majority (*n* = 262; 64.53%) of participants were recruited from schools for the deaf across the country (Supplementary Material Appendix A). Only one administrative region has not been covered during the recruitment. The numbers on the figure correspond to the families recruited in each administrative region.

**Figure 2 genes-14-00562-f002:**
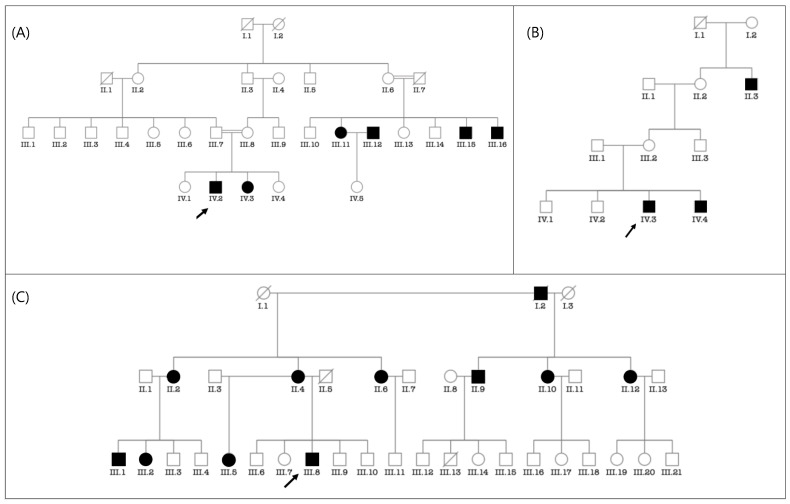
Typical modes of inheritance found in our cohort. (**A**) Pedigree of a consanguineous family with autosomal recessive non-syndromic hearing impairment. (**B**) Pedigree of a family with non-syndromic hearing impairment suggestive of X-linked recessive inheritance. (**C**) Pedigree of a family with autosomal dominant non-syndromic hearing impairment. Probands are indicated by arrows.

**Figure 3 genes-14-00562-f003:**
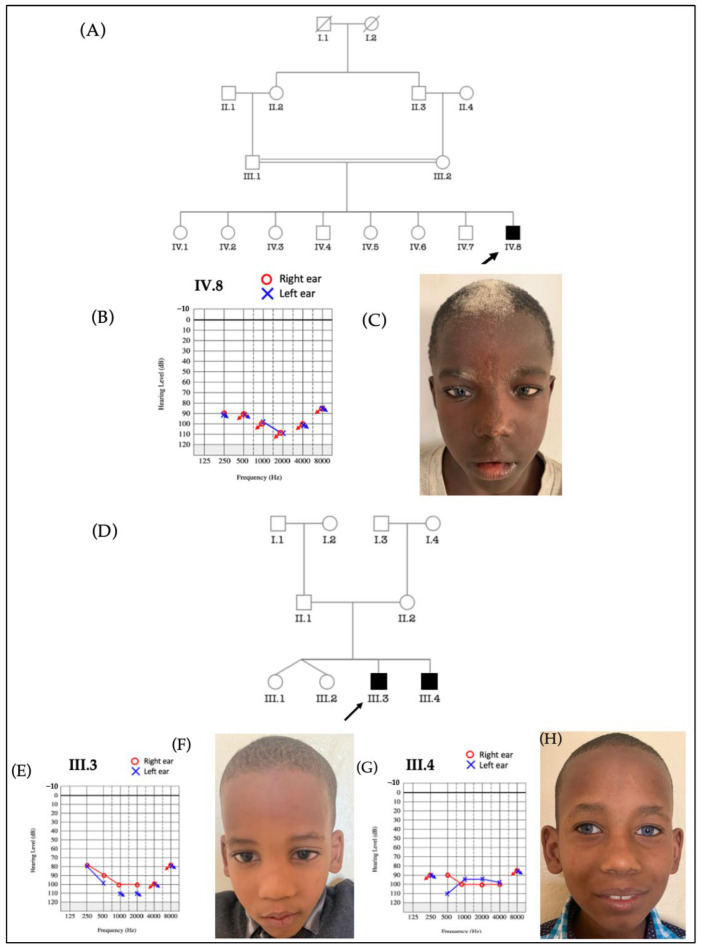
Two families with Waardenburg syndrome type 2. In the first family (**A**) the proband (**C**) exhibited white forelock, sapphire blue eyes, and hypopigmented skin patches associated with profound sensorineural HI (**B**). In the second family (**D**), the proband (**F**) showed light skin, white hair, and profound HI (**E**), while his brother (**H**) exhibited sapphire blue eyes and light skin associated with profound HI (**G**).

**Figure 4 genes-14-00562-f004:**
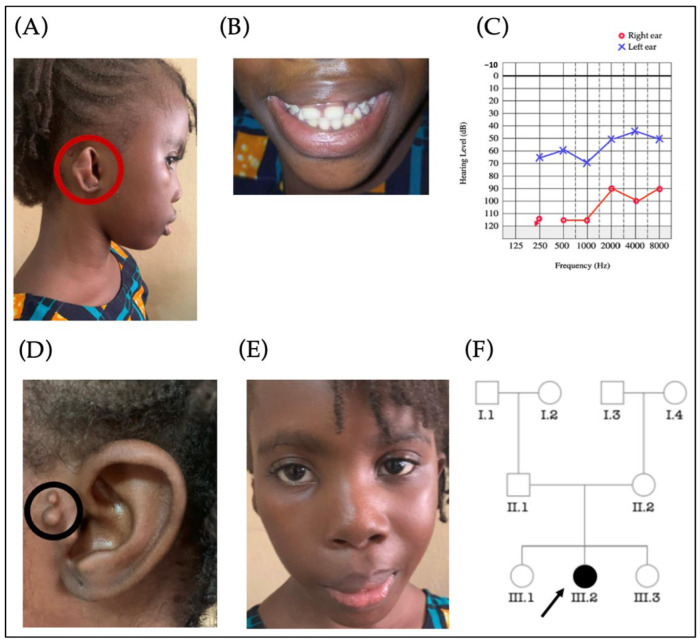
Patient with a suspected Labyrinthine Aplasia, Microtia, and Microdontia syndrome. The photo is illustrating a 7-year-old girl showing unilateral microtia (Type 3) (**A**), microdontia with wide-spaced teeth (**B**), periauricular skin tags (**C**), profound hearing loss in the right ear while the left ear exhibited moderate hearing loss (**D**), and asymmetrical face (**E**). The pedigree (**F**) suggests an autosomal recessive pattern of inheritance. The arrow indicates the proband.

**Figure 5 genes-14-00562-f005:**
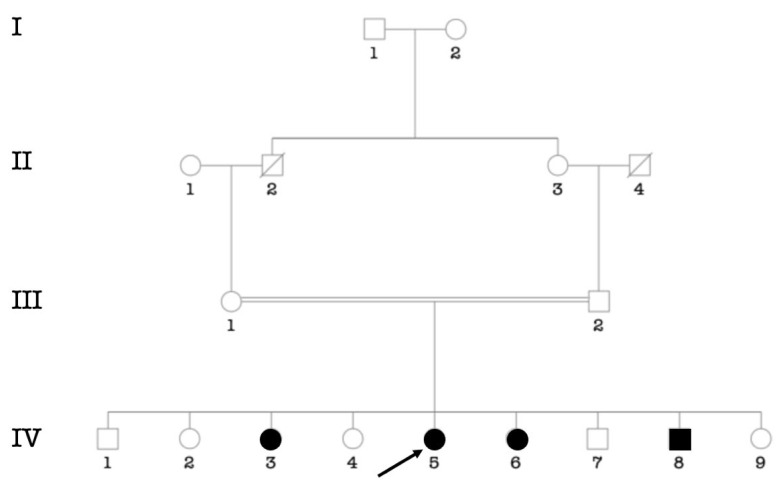
Pedigree showing four affected individuals from a consanguineous marriage segregating Pendred syndrome. The arrow indicates the proband.

**Figure 6 genes-14-00562-f006:**
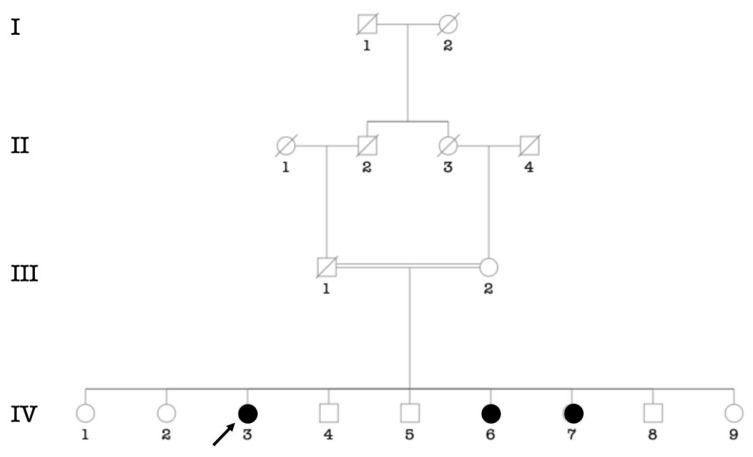
Pedigree of a multiplex family showing three affected girls from a non-consanguineous marriage segregating Diabetes-deafness syndrome. The arrow indicates the proband.

**Figure 7 genes-14-00562-f007:**
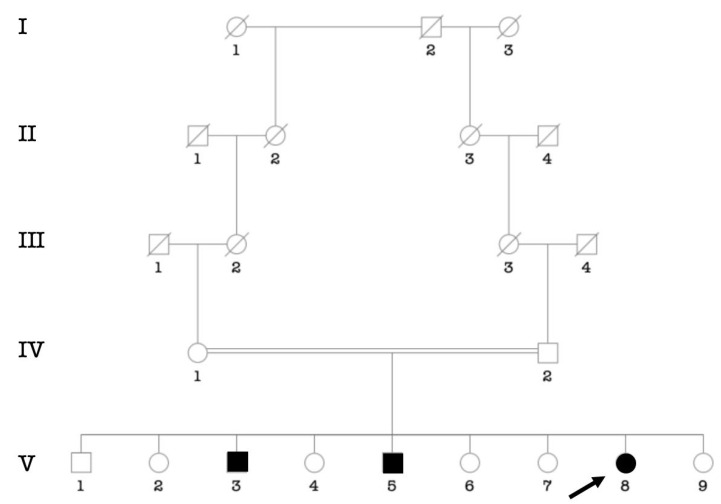
Pedigree showing three affected individuals segregating Usher syndrome. The arrow indicates the proband.

**Figure 8 genes-14-00562-f008:**
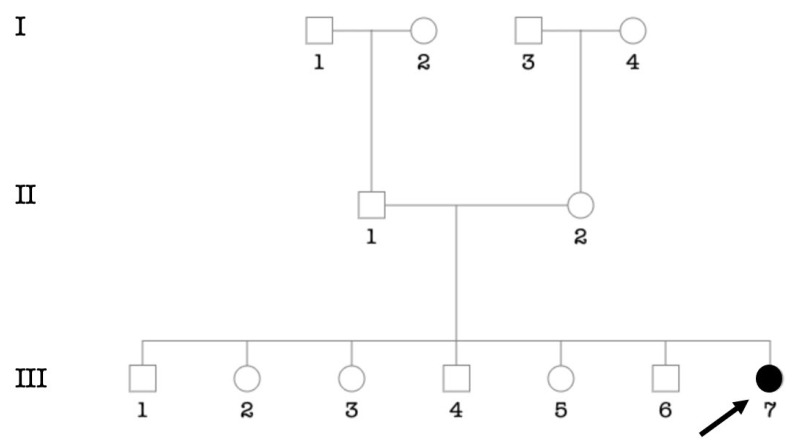
Pedigree of a simplex family with Down’s syndrome. The arrow indicates the proband.

**Figure 9 genes-14-00562-f009:**
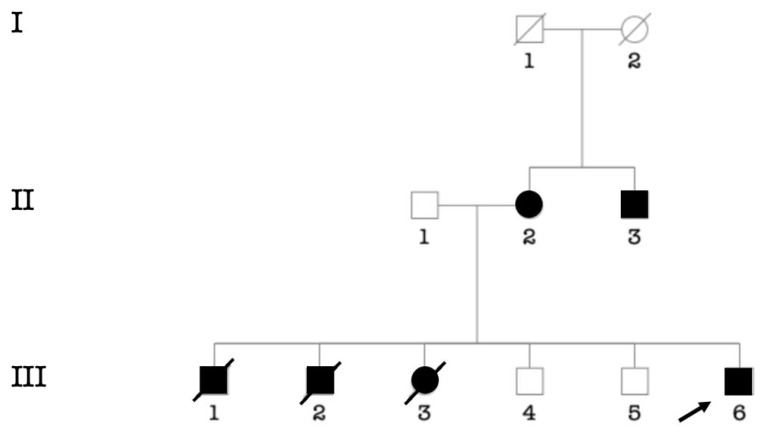
Pedigree showing six affected individuals segregating likely Alport syndrome. The arrow indicates the proband.

**Figure 10 genes-14-00562-f010:**
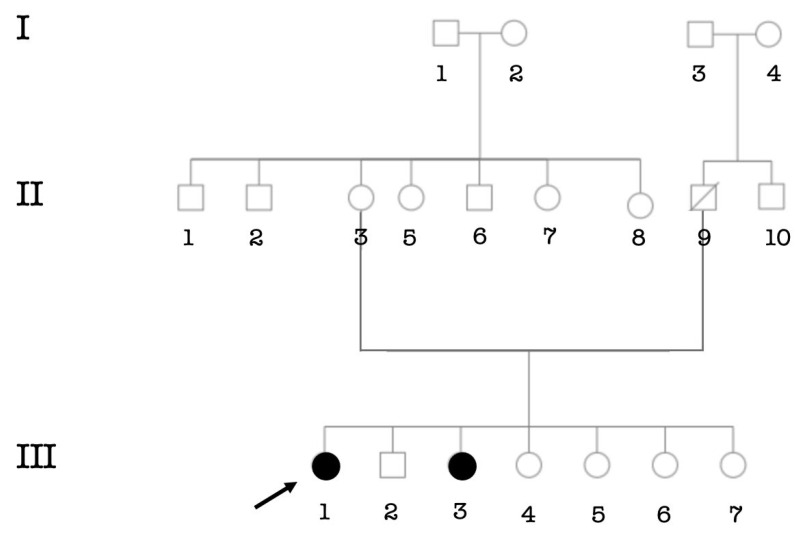
Pedigree showing two affected individuals from a non-consanguineous marriage. The arrow indicates the proband.

**Table 1 genes-14-00562-t001:** Participants’ Sociodemographic and Onset of Hearing Impairment.

Characteristics			N	%
Onset	Prelingual	Before 2 years old	325	80.05
	Perilingual	Between 2 and 4 years old	29	7.14
	Post lingual	After 4 years old	52	12.81
	Age at medical diagnosis	[0–2 years]	159	39.16
		[2–4 years]	112	27.58
		After 4 years	134	33.00
Sociodemographic	Sex	Male	232	57.14
		Female	174	42.86
	Level of education	Preschool	94	23.15
		Primary	180	44.33
		Secondary	13	3.20
		Tertiary	1	0.25
		None	69	17
		Too young to attend a school	49	12.07

**Table 2 genes-14-00562-t002:** Audiometric characteristics and hearing rehabilitation.

Characteristics		*n*	%
Type of HI	Sensorineural	362	89.16
	Conductive	37	9.11
	Mixed	7	1.72
Degree of HI	Moderate (40–60 dB)	65	16.01
	Severe (61–80 dB)	66	16.26
	Profound (≥81 dB)	275	67.73
Symmetry of HI	Symmetric	381	93.84
	Asymmetric	25	6.16
Laterality of HI	Bilateral	401	98.77
	Unilateral	5	1.23
Curve pattern	Flat	301	81.79
	Sloping	45	12.23
	Ascendant	9	2.45
	No Curve (Total hearing loss)	13	3.53
Hearing rehabilitation	Conventional hearing aids	15	3.70
	Cochlear implantation	3	0.74
	Absence of hearing rehabilitation	388	95.56

**Table 3 genes-14-00562-t003:** Etiologies of Childhood Hearing Impairment in Senegal.

Etiologies			*n*	%
Environmental		Chronic otitis media	38 (38 families)	9.36
		Meningitis	25 (25 families)	6.16
		Prematurity	11 (11 families)	2.71
		Head Trauma	9 (9 families)	2.22
		Neonatal asphyxia	9 (9 families)	2.22
		Cerebral malaria	6 (6 families)	1.48
		Neonatal infection	7 (7 families)	1.72
		Neonatal jaundice	2 (2 families)	0.49
		Ototoxicity	2 (2 families)	0.49
		Low birth weight	2 (2 families)	0.49
		Eustachian Tube Dysfunction	2 (2 families)	0.49
		Others *	7 (7 families)	1.72
Likely genetic	Non-Syndromic HI	Autosomal recessive	153 (59 families)	37.69
		Autosomal dominant	7 (3 families)	1.72
		Recessive X-linked	7 (3 families)	1.72
		Sporadic	27 (27 families)	6.65
	Syndromic HI	Waardenburg type 2	5 (4 families)	1.23
		Pendred Syndrome	4 (1 family)	0.99
		Usher type 2 Syndrome	3 (1 family)	0.74
		Diabetes-deafness Syndrome	3 (1 family)	0.74
		LAMM Syndrome	1 (1 family)	0.25
		Down’s Syndrome	1 (1 family)	0.25
		Alport Syndrome	1 (1 family)	0.24
		Non-Specific Syndrome	2 (1 family)	0.49
Unknown			72 (72 families)	17.74
		Total	406 (295 families)	100

* Other = Tympanic perforation, recurrence of earwax, velopharyngeal insufficiency, congenital infection, and malnutrition.

**Table 4 genes-14-00562-t004:** Consanguinity rate among the main ethnic groups.

Ethnic Groups	Number of Families	Consanguinity Rate (%)
Wolof	53	90.6
Fulani	10	100
Serer	21	100
Manding	2	50
Other *	4	75
Total	90	93

Other *: Diola, Manjaque and Mankagne.

**Table 5 genes-14-00562-t005:** Comparison of our results to other studies in the neighboring countries of Senegal.

	Gambia	Mali	Mauritania	Guinea	Present Study
Year of publication	1985	2021	2016	2017	2022
Reference	[24]	[14]	[25]	[26]	--
Number of patients	257	117 (100 families)	139 (113 families)	124	406 (295 families)
Hereditary	8%	29.3%	44.4%	18.5%	52.6%
Consanguinity rate	--	55.5%	61.3%	--	93%
Contribution of GJB2	--	0%	9.4%	--	34.1%
Chronic otitis	--	7.7%	--	--	9.36%
Cerebrospinal meningitis	31%	40%	13%	42%	6.16%
Ototoxic medication	--	14.3%	3.6%	16.1%	0.49%
Cerebral malaria	16%	--	--	--	1.48%
Prematurity	--	15.7%	--	--	2.17%
Head trauma	--	--	--	--	2.22%
Neonatal asphyxia	--	2.8%	--	8.1%	2.22%
Neonatal jaundice	--	--	--	--	0.49%
Measles	2%	--	2.9%	--	--
Rubella	1.5%	--	--	--	--
Unknown	54.4%	11.3%	5.7%	22.6%	17.74

## Data Availability

All datasets supporting the conclusion of this study are included in the Appendix A.

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
