# Peer review of "Childhood Hearing Impairment in Senegal"

_genes, 2023, doi:10.3390/genes14030562_

Round 1
Reviewer 1 Report
This observational study has recruited 406 participants from Senegal to determine the inheritance pattern of hearing loss and they have made some interesting conclusions in terms of epidemiology. It’s a lot of work to recruit large cohort of individuals and investigate the inheritance pattern by pedigree analysis, and hearing loss phenotype. The paper is concise with appropriate analysis supported by well drawn informative figures/tables. However, I would have liked to seen the identification of causative mutations in some of the known genes to see if they differ from previously reported ones.
Corrections:
Introduction - unclear if Senegal has a newborn hearing screen in place or not (assuming no based on the later recommendation in the conclusion). Include a statement on this earlier.
Figure 1 - include a worldwide or African map to indicate where Senegal is.
Author Response
Comments from Reviewer 1
Comments and suggestions for Authors: This observational study has recruited 406 participants from Senegal to determine the inheritance pattern of hearing loss and they have made some interesting conclusions in terms of epidemiology. It’s a lot of work to recruit large cohort of individuals and investigate the inheritance pattern by pedigree analysis, and hearing loss phenotype. The paper is concise with appropriate analysis supported by well-drawn informative figures/tables. However, I would have liked to seen the identification of causative mutations in some of the known genes to see if they differ from previously reported ones.
Authors’ Response: Thank you for your appreciative words. In this paper we wanted to focus on a comprehensive description of clinical features of HI across the Senegal which, as the reviewer kindly noted, is a considerable endeavor. As stated in the text among the 44 multiplex families which are part of this work, we investigated the contribution of GJB2. We showed that variants in GJB2 explained 34% of multiplex families (https://doi.org/10.3390/biology11050795) (Page 2, line 84). For GJB2 negative families, we are currently performing Whole Exome Sequencing. We hope that the reviewer will understand that this WES analysis will we reported as a separate manuscript, when available.
Corrections: Introduction - unclear if Senegal has a newborn hearing screen in place or not (assuming no based on the later recommendation in the conclusion). Include a statement on this earlier.
Authors’ response: We agree with your assessment. We have now stated in the introduction that: “There is not yet universal newborn hearing screening program in Senegal” (Page 2, line 83).
Figure 1- include a worldwide or African map to indicate where Senegal is.
Authors’ response: Thanks for the suggestion, that is now incorporated in the revised figure 1.
Reviewer 2 Report
1. The Methods section stated that: "In the other administrative regions, we recruited participants from the schools for the 119 deaf and/or in the community." However, later in the text, it was shown that there was only one school for the Deaf in the whole country and that school is in Dakar. It is pertinent to clearly state that recruitment in other regions outside Dakar was by community recruitment.
2. Community recruitment through a single individual carries inherent risk of bias. This limitation must be clearly stated.
3. There are many authorities who believe that satisfactory output from Pure Tone Audiometry (PTA) in children is guaranteed from age 5years and above. Reliable results in younger children is dependent on the concentration level of each child. Thus the decision of the authors to conduct PTA for all participants from age 2years is very concerning! Further explanation is required to justify this approach otherwise there is risk of false results.
4. Minor spelling error in Table 1. Please see: "Age at medical diagnostic". It should read "Age at medical diagnosis"
5. Line 196 and Table 3, "Chronic otitis" does not convey any clinical interpretation. Otitis simply means inflammation of the ear. But the ear is divided into outer, middle and inner ear. Please correct to indicate which part of the ear is inflamed, though I suspect the author meant chronic otitis media, it is pertinent that the right clinical terminologies are used.
6. Table 3. "Rhino-pharyngitis" without conjunction with additional etiology such as Eustachian Tube Dysfunction is NOT a known cause of hearing loss. Presenting inaccurate clinical information may cast aspersion on this manuscript.
7. Line 323. Comparison of this study's result to another study done in Gambia in 1985 requires a little stretch of imagination. Investigations such as OAE,ABR and even a good measure of clinical knowledge available now what not so in 1985. It is appropriate to either state the known discrepancies or drop the Gambian reference
Author Response
Comments from Reviewer 2
- The Methods section stated that: "In the other administrative regions, we recruited participants from the schools for the 119 deaf and/or in the community." However, later in the text, it was shown that there was only one school for the Deaf in the whole country and that school is in Dakar. It is pertinent to clearly state that recruitment in other regions outside Dakar was by community recruitment.
Authors’ response: Thank you for pointing this out. We have revised as follows:
“There is only one public school for the deaf in the country, i.e. Centre Verbo-Tonal (CVT) de Dakar. The schools for the deaf are not funded by the Senegalese government. Therefore, outside Dakar, participants were recruited from the non-public schools as well as through community engagement” (page 3. Line 116-118)
- Community recruitment through a single individual carries inherent risk of bias. This limitation must be clearly stated.
Authors’ response: Thank you for raising this point. We have clarified and revised the text as follows (page 3, lines 116-119): “In most region, we started our recruitment with one community leader. However, over the course of the recruitment process, we identified additional community leaders who actively participated in the process of identifying families. Because these community leaders were the pillars of community-based recruitment, in absence of one the Sédhiou administrative region, we could not perform any recruitment in that region (Figure 1)”.
- There are many authorities who believe that satisfactory output from Pure Tone Audiometry (PTA) in children is guaranteed from age 5years and above. Reliable results in younger children are dependent on the concentration level of each child. Thus, the decision of the authors to conduct PTA for all participants from age 2years is very concerning! Further explanation is required to justify this approach otherwise there is risk of false results.
Authors' response: Thank you for bringing this to our attention. We have included the following in the discussion (Page 17, line 487-493):
“A possible limitation of this study was the use of Pure Tone Audiometry (PTA) to assess hearing impairment for preschool’s children aged 2-5 years, because this was the only approach available to us in most setting. Nevertheless, to alleviate this challenge, we performed behavioral audiometry using the same equipment as in children over 5 years, with an adaptation of the technique (conditioning with toys), despite this being time consuming. Depending on the child's level of concentration for some children, the test was conducted over several days in order to obtain a reliable result.”
- Minor spelling error in Table 1. Please see: "Age at medical diagnostic". It should read "Age at medical diagnosis"
Authors' response: Thank you. This was revised (page 4, line 187).
- Line 196 and Table 3, "Chronic otitis" does not convey any clinical interpretation. Otitis simply means inflammation of the ear. But the ear is divided into outer, middle and inner ear. Please correct to indicate which part of the ear is inflamed, though I suspect the author meant chronic otitis media, it is pertinent that the right clinical terminologies are used.
Authors' response: Thank you for this remark, we omitted “media” (page 7, line 258). This is addressed accordingly.
- Table 3. "Rhino-pharyngitis" without conjunction with additional etiology such as Eustachian Tube Dysfunction is NOT a known cause of hearing loss. Presenting inaccurate clinical information may cast aspersion on this manuscript.
Authors' response: You have raised an important point. we have revise to Eustachian Tube Dysfunction (that was associated with eardrum retraction among two of our patients) (page 7, line 258).
- Line 323. Comparison of this study's result to another study done in Gambia in 1985 requires a little stretch of imagination. Investigations such as OAE, ABR and even a good measure of clinical knowledge available now what not so in 1985. It is appropriate to either state the known discrepancies or drop the Gambian reference
Authors' response: It is true that the Gambian reference dates back almost four decades, but we have not found a more recent publication on the causes of childhood hearing impairment in Gambia, and we hope than the reviewer will understand that we elected to keep this reference, despite the limitation of its interpretation. Please note that we did compare the causes of childhood hearing impairment in our study to other studies in the neighboring countries of Senegal, as well!
Round 2
Reviewer 1 Report
The feedback from the reviewers has been incorporated into the current draft or a totally reasonable justification. The corrections have added to the quality of the paper and improve the interest to the readers. In particular, recruitment of subjects in an African country with cultural appropriateness and distribution of affected individuals with hearing loss.
Author Response
Many thanks for your positive comments:
The present paper describes a comprehensive clinical profile of HI in an African country, i.e., Senegal, which will add to the scares and needed data from that region.
We have now added the above comment in the text (lines 500-502).